# Human-Understandable Explanations of Infeasibility for Resource-Constrained Scheduling Problems

**Niklas Lauffer, Ufuk Topcu** [*]
{nlauffer, utopcu}@utexas.edu
University of Texas, Austin

## Abstract

Significant work has been dedicated to developing methods for communicating reasons for decision-making within automated scheduling and planning systems to human users. However, much less focus has been placed on communicating reasons for why scheduling systems are unable to arrive at a feasible solution when over-constrained. We investigate this problem in the context of task scheduling. We introduce the agent resource-constrained project scheduling problem (ARCPSP), an extension of the resource-constrained project scheduling problem which includes a conception of agents that execute tasks in parallel. We outline a generic framework, based on efficiently enumerating minimal unsatisfiable sets (MUS) and maximal satisfiable sets (MSS), to produce small descriptions of the source of infeasibility. These descriptions are supplemented with potential relaxations that would fix the infeasibility found within the problem instance. We illustrate how this method may be applied to the ARCPSP and demonstrate how to generate different types of explanations for an over-constrained instance of the ARCPSP.

## 1  Introduction

In many real-world applications, human users in charge of developing plans and making decisions are aided by automated planning and scheduling systems. For example, NASA mission planning makes use of a large team of human planners that use various automated scheduling systems in order to construct day-to-day as well as long-term plans for crew members. A primary function of these automated systems is generating different types of plans and schedules while ensuring that various constraints do not conflict. When plans are ultimately constructed by human planners for a human crew, it is essential for both the planners, and the crew executing the plans, to understand how and why certain scheduling decisions were made by automated tools. In general, when the primary function of such constraint satisfaction and optimization tools is to support human decision-making, it is necessary for the automated systems to be transparent in how they arrive at certain outputs.

Significant work has been dedicated to generating human-understandable explanations for why certain automated planning decisions were made (Seegebarth et al. 2013).

[*]This work has been supported in part by the grants NASA NNX17AD04G and NSF 1652113.

However, little work has been done in generating reasons for why plans or schedules *cannot* be generated under certain specifications. Human users interacting with such constraint satisfaction or optimization tools are bound to run into configurations for which no feasible solution exists. Fixing infeasible configurations is a challenging task for the human user if they are unable to understand why the solver arrives at an unsatisfiable conclusion.

While various partial constraint satisfaction tools exist for solving such over-constrained problems (Freuder and Wallace 1996), solutions employing these tools have significant limitations that make them less applicable in certain real-life scenarios. Most of these methods employ constraint hierarchies to determine which constraints should be violated in order to satisfy more important ones. However, in complicated planning or scheduling applications involving multiple human agents, constructing such a hierarchy is often impractical. Instead, if reasons for infeasibility can be properly conveyed back to the human user, they can make high-level decisions to solve infeasibility in any way they see fit.

In this paper, we provide a framework for iteratively generating human-understandable *explanations* of infeasibility for a specific class of scheduling problems. These explanations manifest themselves as minimal sets of specifications (or constraints) that are responsible for causing infeasibility, coupled with suggestions for relaxations through which feasibility could be achieved.

The method proposed in this paper allows users to enumerate over a series of explanations for infeasible instances of problems at varying levels of abstraction. For example, raw explanations of relevant low-level constraints may be directly output or a causal link may be established back to higher level descriptions of the problem to understand what specifications were responsible for the feasibility issue. This system also allows directed questions about feasibility to be asked, such as "why can task A not be scheduled after task B?"

A strategy for iteratively generating minimal unsatisfiable sets (MUS) and maximal satisfiable sets (MSS) forms the basis for interpreting the infeasibility of the problem. Existing methods such as QuickXplain (Junker 2004) focus on generating a single most preferable explanation of infeasibility. Likewise, (Burt, Klimova, and Primas 2018) aims to generate a single explanation in the context of optimization

without attempting to achieve minimality. However, over-constrained problems may contain several infeasibility issues which cannot be solved by changing only a single part of the problem. So, because a single MUS only provides indication of a single feasibility issue, we aim to enumerate several sets of MUS to highlight multiple feasibility issues found within the problem instance. Therefore, the proposed enumeration strategy is based on MARCO (Liffiton et al. 2016), a flexible algorithm for generating MUSes and MSSes in succession.

Motivated by the domain of space mission scheduling, we introduce and investigate the agent resource-constrained project scheduling problem (ARCPSP), an extension of the resource-constrained project scheduling problem (RCPSP) that incorporates the delegation of tasks to differing agents. This problem cannot be framed as an instance of the RCPSP because it deals with the case of asymmetric agents in which certain tasks may only be executed by a subset of the agents. This problem is meant to model applications in which efficient scheduling for teams of differing agents is critical. While we only explicitly investigate this problem, the generality of the approach outlined in this paper would allow the methodology to be adapted for different types of constraint satisfaction and optimization tools as well as different types of planning and scheduling problems.

The main contributions of this paper are the following: firstly, we provide a formal definition of the agent resource-constrained project scheduling problem (ARCPSP) in Section 3. Then in Section 4 we outline a difference logic encoding of the ARCPSP which is used to check feasibility of problem instances. The framework for generating human-understandable explanations of infeasibility for instances of the ARCPSP is described in Section 5. Finally, we provide an overview of the trade-off between interpretability and expressibility of different types of explanations and conclude by discussing how these ideas can be extended.

## 2  Preliminaries and Definitions

In this section, we introduce relevant background information and definitions used throughout the paper. These concepts will set the stage for formulating the ARCPSP in terms of satisfiability modulo theory and using minimal unsatisfiable sets and maximal satisfiable sets to generate explanations of infeasibility.

### 2.1  Boolean Satisfiability

Let $X$ be a set of variables and clauses $C_1, \ldots, C_n$ be formulas representing constraints over $X$. Consider a formula of the form

$$\varphi = \bigwedge_{i=1,\ldots,n} C_i. \tag{1}$$

We say the formula $\varphi$ is *satisfiable* if there exists some assignment to the variables in $X$ which makes $\varphi$ evaluate to TRUE. Otherwise, it is *unsatisfiable*. Note that if $\varphi$ takes the form of equation (1), as it does throughout this paper, every clause $C_i$ must be TRUE in order for $\varphi$ to evaluate to TRUE. To implement the temporal constraints within a schedule, the clauses $C_i$ are taken from the theory of *difference logic*

(DL), which makes deciding $\varphi$ a satisfiability modulo theory (SMT) problem. To check satisfiability of problem instances, we use the Microsoft Z3 SMT solver (De Moura and Bjørner 2008).

### 2.2  Difference Logic

As will be discussed in Section 4, the agent resource-constrained project scheduling problem (ARCPSP) can be encoded in *difference logic* (DL), a fragment of linear real arithmetic (LRA). The numerical components of DL are solvable in polynomial time (Cotton and Maler 2006) using graph-based procedures based on an incremental Bellman-Ford algorithm. In general, decidability for DL using these methods is more efficient than the simplex-based methods used to decide LRA. Under DL, atoms are restricted to the form

$$x - y \le k \quad \text{for} \quad x, y \in X,\ k \in \mathbb{R}. \tag{2}$$

However, we can rewrite the following atoms in difference form:

- $x - y \ge k \quad \equiv \quad (y - x \le -k)$
- $x - y = k \quad \equiv \quad (x - y \le k) \wedge (x - y \ge k)$
- $x = y \quad \equiv \quad x - y = 0$

Bounds $x \le k$ can also be incorporated by writing them as $x - x_0 \le k$ where $x_0$ is a special variable that is later set to zero.

### 2.3  Minimal Unsatisfiable and Maximal Satisfiable Sets

**Definition 1.** A *minimal unsatisfiable set* (MUS) of a set $C$ of constraints is a subset $M \subseteq C$ such that $M$ is unsatisfiable and every proper subset $M' \subset M$ is satisfiable. A *maximal satisfiable set* (MSS) of a set $C$ of constraints is a subset $M \subseteq C$ such that $M$ is satisfiable and every proper superset $M'$, with $C \supseteq M' \supset M$, is unsatisfiable. A *minimal correction set* (MCS) of a set $C$ of constraints is the complement of some maximal satisfiable set of $C$, and can be understood as a minimal set of constraints which need to be removed from $C$ in order to make it satisfiable.

It is important to note that MUSes, MSSes, and MCSes are only *locally* maximal (or minimal), and are different from concepts of globally optimal subsets. MUSes can be understood as isolated, infeasible subsets of the constraints. Their primary characteristic is that removing any single constraint would make the set satisfiable. However, this does not necessarily guarantee the feasibility of the entire set of constraints because there might be many disjoint MUSes within the set. In order to make the entire set feasible (satisfiable), a *hitting set* of the MUSes must be removed. Every MCS is precisely one combination of such a hitting set.

**Definition 2.** A *background* of a set $C$ of constraints is a subset $B \subseteq C$ of hard constraints, which must be necessarily satisfied. In the context of scheduling problems, backgrounds typically include constraints that ensure that the outcome of the schedule is logical, including conditions such as tasks not overlapping and resource constraints not being exceeded. We denote everything outside of the background

$M \setminus B$ as the *foreground*. Hence, the background and foreground partition the set $C$ of constraints.

A *minimal conflict* of an over-constrained set $C$ of constraints with respect to a background $B$ is then a subset of the foreground $M \subset C \setminus B$ such that $M \cup B$ is unsatisfiable and, for any superset $M' \supset M$, $M' \cup B$ is satisfiable. A *minimal relaxation* of an over-constrained set $C$ of constraints with respect to a background $B$ is a subset of the foreground $M \subset C \setminus B$ such that $(C \setminus M) \cup B$ is satisfiable and, for any superset $M' \supset M$, $(C \setminus M') \cup B$ is unsatisfiable. Then an *explanation* is a sequence of minimal conflicts and minimal relaxations for a problem instance.

The definitions of minimal conflicts and minimal relaxations mirror the concepts of MUSes and MCSes, respectively, while incorporating a background of constraints which cannot be modified. A background is necessary for specifying hard constraints which cannot be relaxed or modified. This way we can prevent certain constraint from consideration for conflicts or relaxations. A background also allows the generation of explanations concerning different aspects of a scheduling problem instance, a concept which will be explored later in the paper.

## 3 Problem Description

The problem that we formulate is an extension of the resource-constrained project scheduling problem (RCPSP). Loosely, the RCPSP considers nonpreemptive, precedence-constrained tasks of known durations that are constrained by reusable resource requirements (i.e. resources that are returned after a task stops using them). The agent resource-constrained project scheduling problem extends the RCPSP to include a set number of agents that execute the tasks in parallel, subject to certain compatibility constraints. Additionally, while the RCPSP generally cares about optimizing the total makespan of the schedule, we instead introduce a set start and end time for each scheduling instance and only focus on its feasibility (i.e. whether or not all tasks can be completed within this specified time frame).

### 3.1 The Agent Resource-Constrained Project Scheduling Problem

An instance of an agent resource-constrained project scheduling problem (ARCPSP) is defined by a tuple $(M, J, s, p, U, E, R, B, b)$, where the components are defined as follows.

– $M = \{M_1, M_2, \cdots, M_m\}$ is a set of agents.

– $J = \{J_1, J_2, \cdots, J_n\}$ is a set of non-preemptive (uninterruptible) tasks.

– $s = [(a_1, b_1), (a_2, b_2), \cdots, (a_n, b_n)]$ are the allowable time ranges in which the tasks should be executed, where $a_i, b_i \in \mathbb{N}$.

– $p = [p_1, \ldots, p_n]$ is a vector of the durations of tasks $J$, where $p_i$ is the duration of task $J_i$.

– $U = \{U_1, U_2, \cdots, U_n\}$ is the compatibility set for the tasks. Each task $J_i$ can be completed by a subset $U_i \subseteq M$ of agent.

– $E \subseteq J \times J$ is a set of precedence relations. $(J_i, J_j) \in E$ if and only if task $J_i$ must terminate before task $J_j$ begins. Precedence relations must be defined in a consistent way (by respecting the transitive property).

– $R = \{R_1, R_2, \cdots, R_q\}$ is a set of reusable resources.

– $B \in \mathbb{N}^q$ represents the total availability of the resources $R$. The tasks that share resource $B_i$ are *mutually exclusive* if $B_i = 1$.

– $b \in \mathbb{N}^{n \times q}$ represents the resource demands of tasks where task $J_i$ requires $b_{i,j}$ units of resource $R_j$ during its execution. The total demand of resource $R_j$ at anytime cannot exceed its total availability $B_j$.

A *schedule* $(S, A) = (\{S_1, S_2, \cdots, S_n\}, \{A_1, A_2, \cdots, A_n\})$ is a solution to an instance of an ARCPSP, where $S_i$ and $A_i$ are the start time and the assigned agent of task $J_i$, respectively. A schedule is *feasible* if it satisfies the following constraints:

• No agent has overlapping tasks,

$$\left(S_i + p_i \leq S_j\right) \vee \left(S_j + p_j \leq S_i\right) \quad (3)$$

$\forall i, j \in (1, \ldots, n)$ such that $A_i = A_j$.

• Every task falls within its allowable time frame

$$\left(s_{i,1} \leq S_i\right) \wedge \left(S_i + p_i \leq s_{i,2}\right) \quad \forall S_i \in S. \quad (4)$$

• The activities are assigned to compatible agents

$$M_i \in U_i \quad \forall M_i \in A. \quad (5)$$

• The precedence relations are met

$$S_i + p_i \leq S_j \quad \forall(J_i, J_j) \in E. \quad (6)$$

• The resource constraints are satisfied, let $\mathcal{J}_t = \{J_i \in J \mid S_i \leq t < S_i + p_i\}$ represent the set of tasks being executed at time $t$, then

$$\sum_{J_i \in \mathcal{J}_t} b_{i,j} \leq Bj \quad \forall R_j \in R, \forall t \geq 0. \quad (7)$$

## 4 SMT Formulation

The constraints of the ARCPSP can be formulated in terms difference logic in the following way. Constraint (3) can be rewritten as

$$(S_i - S_j \leq -p_i) \vee (S_j - S_i \leq -p_j) \vee \neg(A_i = A_j) \quad (8)$$

$\forall S_i, S_j \in S$. Constraint (4) can be rewritten as

$$(s_{i,1} \leq S_i) \wedge (S_i \leq s_{i,2} - p_i) \quad \forall S_i \in S, \quad (9)$$

and constraint (6) can be rewritten as

$$S_i - S_j \leq p_i \quad \forall(J_i, J_j) \in E. \quad (10)$$

By representing the agents as integers $M_i \in \mathbb{N}$, constraint (5) can be rewritten as

$$\bigvee_{u \in U_i} M_i = u \quad \forall M_i \in A \quad (11)$$

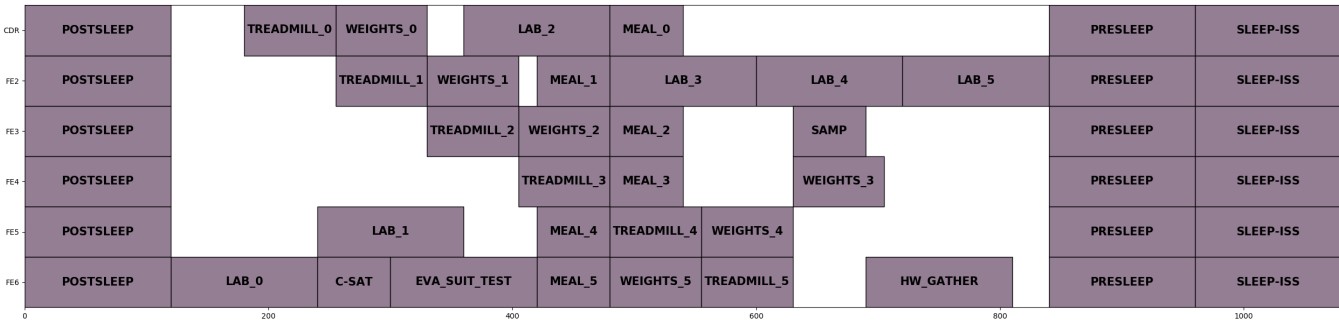

Figure 1: A feasible solution to Example 1.

Encoding the resource constraints is slightly more challenging. For mutually exclusive constraints ($B_i = 1$), the tasks that share a resource can simply be encoded as not being allowed to be executed at the same time. That is, constraint (7) can be rewritten as

$$(S_i - S_j \leq -p_i) \vee (S_j - S_i \leq -p_j) \qquad (12)$$

$\forall b_{i,k}, b_{j,k} = 1, \forall R_k \in R$. We can generalize this idea to non-mutually exclusive constraints through the concept of *minimal forbidden sets*. First introduced by (Möhring and Stork 2009), forbidden sets are unsatisfiable sets with respect to resource constraints only. They represent the sets of tasks that cannot be simultaneously scheduled because they would otherwise exceed the availability of some resource constraint. The essential feature of a minimal forbidden set is that a single task can be rescheduled to another time to make the set respect the resource constraint.

Therefore, given a minimal forbidden set $J^*$, we would like to encode a constraint requiring that they cannot all be executing at the same time

$$\bigvee_{J_i \in J^*} \neg(J_i \in \mathcal{J}_t) \quad \forall t \geq 0 \qquad (13)$$

where $\mathcal{J}_t = \{J_i \in J \mid S_i \leq t < S_i + p_i\}$ represents the set of tasks being executed at time $t$. This encoding is similar to methods which encode the RCPSP in terms of linear arithmetic (Bofill et al. 2016), but this requires discretizing time and incurs a cumbersome number of constraints if there are a large number of time-points. Moreover, equation (13) cannot easily be formulated in terms of difference logic. Instead, we can reformulate the constraint as there being at least two tasks in $J^*$ that do not overlap

$$\bigvee_{J_i \in J^*} \bigvee_{J_j \in J^*} (J_i + p_i \leq J_j) \vee (J_j + p_i \leq J_i) \qquad (14)$$

for every minimal forbidden set $J^*$. This constraint is logically equivalent to requiring that at any time-point, there be at least one task in each minimal forbidden set that is not being executed.

Constraining all of the *minimal forbidden sets*, a subset of all of the forbidden sets, is sufficient to prevent resource conflicts because every forbidden set is a superset of some minimal forbidden set. Algorithms exist for computing all

minimal forbidden sets (Stork and Uetz 2005) so we will not discuss such a computation here.

Encoding resource constraints as forbidden sets is efficient in the context of the ARCPSP in comparison to other methods, such as equation (13). This is primarily because of the computational advantage achieved by difference logic over other theories such as linear real arithmetic and the encoding not requiring a discretization of time. Representing resource constraints as minimal forbidden sets also provides an explicit representation of resource constraints in terms of MUSes. If a resource constraint appears in an explanation, we can represent it as the minimal forbidden set which is being violated. For example, if a resource constraint constitutes a component of some MUS, it will be represented as some subset $\{A, B, C\}$ of tasks, meaning that tasks $A, B,$ and $C$ cannot be scheduled at the same time because they would violate a resource constraint.

**Example 1.** *Scheduling astronauts aboard the ISS*

We model the problem of scheduling astronauts aboard the International Space Station (ISS) as an instance of the ARCPSP for which the elements of $M = \{M_1, M_2, \cdots, M_m\}$ represent the crew members. We consider the case of $m = 6$ astronauts. The bounds on task execution are from minute 120 to 840; the sleeping related tasks outside of this bound are fixed so are not a part of the problem instance. The availability of the *power* resource, is 1000 units. The tasks are divided into different categories:

- There are 6 laboratory tasks, each of duration 120 with allowable time ranges of $(120, 840)$, the entire work day. However, they have precedence constraints $\{(J_{L_i}, J_{L_{i+1}}) \mid 1 \leq i < 6\}$, each laboratory task must be completed before the next one begins. Each laboratory task can be completed by any astronaut, so the compatibility set is all of the astronauts. Each laboratory task also has a power requirement of 400 units.

- There are $m$ weights and $m$ treadmill tasks, one for each agent, each of duration 75. They have allowable time ranges of $(180, 720)$. Each weight and treadmill task must be completed by a unique astronaut so their compatibility sets can be specified by letting the $ith$ task only be completed by astronaut $M_i$. However, there is only one set of weights and treadmill equipment, so we can define reusable resources $R_W$ and $R_T$ both with availability

$B_W$, $B_T = 1$, respectively. Each treadmill task also has a power requirement of 200 units.

– There are $m$ meal tasks. They have allowable time ranges of $(420, 540)$. Similar to the exercise tasks, each one must be completed by a unique astronaut so their compatibility sets can be specified by letting the $ith$ meal task only be completed by astronaut $M_i$.

– Several miscellaneous tasks, *deploy_cubesat*, *collect_sample*, *hardware_gather*, and *eva_suit_test* with durations 60, 60, 120, and 120 respectively, do not fall into any particular group. These tasks have an allowable time range of $(120, 840)$ and can all be completed by any astronaut. They require 400, 500, 400, 400 units of power, respectively.

A feasible schedule for this instance of the ARCPSP is visualized in Figure 1.

**Example 2.** *An Infeasible Modification*

We modify the previous example slightly to produce an unsatisfiable problem instance. If we change the duration of each laboratory task from 120 to 121, we get an over-constrained system of constraints for which no feasible schedule exists. We'll use this running example to produce explanations in the next several sections.

# 5   Subset Enumeration: Finding Conflicts and Relaxations

The proposed strategy for enumerating subsets is based on MARCO (Liffiton et al. 2016) over other systems such as CAMUS (Liffiton and Sakallah 2008) because outputting at least some MUSes quickly is more important than explicitly generating every MUS. MARCO relies on four main functions to explore the feasibility of a power set lattice of constraints which we outline here in the language of conflicts and relaxations.

– **BlockUp** - Called whenever a conflict is found. Marks every supersets of the current set, preventing it from being explored later on.

– **BlockDown** - Called whenever a relaxation is found. Marks every subsets of the current set, preventing it from being explored later on.

– **Grow** - If the current set is satisfiable, adds constraints to the current set until adding any other constraint would make it unsatisfiable.

– **Shrink** - If the current set is unsatisfiable, removes constraints from the current set until removing any other constraint would make it satisfiable.

A power set lattice of Boolean variables representing each constraint in the foreground is maintained throughout the execution of the algorithm. First, a random subset of constraints is constructed by choosing a point in the Boolean lattice. Then the SAT solver checks whether the set is `SAT` or `UNSAT`. If the resulting assignment is `SAT` (feasible), then constraints are iteratively added to the current set until a minimal relaxation is found. If the initial set is instead

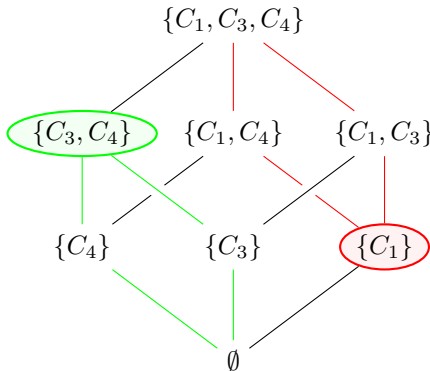

Figure 2: The power set lattice of $\{C_1, C_3, C_4\}$ with background $C_2$ and corresponding relaxation (green) and conflict (red).

`UNSAT` (infeasible), constraints are iteratively removed until a minimal conflict is found. After a minimal conflict is found, **BlockUp** is called, removing any supersets of the minimal conflict from consideration in the lattice. We can do this because any superset of a conflict must be unsatisfiable because it contains the conflict. The opposite direction also applies, any subset, after a minimal relaxation has been removed, must be satisfiable so we can rule them out of consideration. Hence, after a minimal relaxation is found, we **BlockDown**, removing any subsets from consideration in the lattice. Then a new satisfying assignment is pulled from the remaining sections of the boolean lattice and new conflicts and relaxations are generated until the entire lattice is blocked or the algorithm times out.

**Example 3.** *A Small Over-Constrained Formula*

Consider the unsatisfiable conjunction of the following set of clauses,

$$C_1 = \{a\}, \ C_2 = \{\neg a\}, \ C_3 = \{\neg a \vee b\}, \ C_4 = \{\neg b\}$$

with background $B = \{C_2\}$. We'll use this example to step through an execution of the subset enumeration algorithm. A visualization of the associated Boolean lattice is shown in Figure 2. A random initial seed has us start with clause $\{C_1, C_3\}$ and the SAT solver says it's `UNSAT`. We then **Shrink** and remove $C_3$ from the set and the SAT solver says $\{C_1\}$ is still `UNSAT` and minimal. We then output this minimal conflict $\{C_1\}$. Because this set is now minimal, we can **BlockUp**, removing supersets $\{C_1\}$, $\{C_1, C_3\}$, $\{C_1, C_4\}$, $\{C_1, C_3, C_4\}$ from consideration. We then choose a new seed, let's say $\{C_3\}$. The resulting set is `SAT` so we **Grow** to the set $\{C_3, C_4\}$ which is then `SAT` and maximal so we can **BlockDown** subsets $\{C_4\}$, $\{C_3\}$, $\{\emptyset\}$ and output $\{C_1\}$, the complement of $\{C_3, C_4\}$, as a relaxation. The lattice is then entirely blocked off so we terminate with the single conflict and relaxation. Figure 2 shows the power set lattice of the foreground along with the corresponding relaxation (green) and conflict (red).

## 5.1 Background Constraints for the ARCPSP

The standard background for the ARCPSP involves constraints to ensure that the resulting schedule is logical. This way, the foreground only involves constraints which can be altered by parameters that are controlled by the user.

- New variables $S_0$ and $S_{m+1}$ are introduced that mark the beginning and end of the schedule bounds. These variables prevents the subset enumeration from relaxing the temporal constraints of tasks outside of the feasible bounds of the schedule

$$(S_0 \leq S_i) \wedge (S_i + p_i \leq S_{n+1}) \quad \forall S_i \in S. \quad (15)$$

- Each task is assigned to *some* existing agent. Without this constraint, the solver could assign tasks to a nonexistent agent to solve conflicts

$$M_i \in M \quad \forall M_i \in A. \quad (16)$$

- No agent has overlapping activities. Disallowing the solver from consider cases in which tasks can overlap prevents it from generating meaningless results. This condition is precisely constraint (3).

We will refer to this background set of constraints throughout the following section.

## 5.2 Constraint Explanations

The method of generating minimal conflicts and relaxations can be applied to both sets of individual constraints and, by modifying the background, sets of tasks. In this section, we investigate the first case, which we call *constraint explanations*. Following the strategy in the beginning of Section 5, we enumerate only over the constraints that are in the foreground, as specified in Section 5.1. That is, we consider constraints `timeframe`, `compatibility`, `precedence`, and `resource` referring to equations (4), (5), (6), and (7), respectively, for each task in the schedule. The rest of the constraints are implied as a part of the background because they only correspond to imposing a logical structure on the solution, not constraining the parameters of the schedule. Hence, the Boolean lattice which is enumerated over contains only these four types of constraints.

The outputs for constraint explanations are formatted as a tuple of the relevant tasks followed by a constraint type. For example, `(LAB_0, LAB_1)_precedence` refers to the precedence constraint between the first and second laboratory tasks. When a constraint is only relevant to a single task, we write the task followed by the constraint type (e.g., `MEAL_0_compatible` refers to the agent compatibility constraint for the first meal task).

The full constraint explanation for Example 2 includes 14 minimal relaxations and 3 minimal conflicts. Figure 3 shows a representative part of this full constraint explanation. The omitted conflicts and relaxations are identical in structure to the ones shown in Figure 3 and give practically redundant information. Computing the set of minimal forbidden sets took 2.11 seconds and calculating the full explanation took 1.57 seconds.

| | Constraints |
|---|---|
| Confl 1 | $\{$`(LAB_0,LAB_1)_precedence,` `(LAB_1,LAB_2)_precedence,` `(LAB_2,LAB_3)_precedence,` `(LAB_3,LAB_4)_precedence,` `(LAB_4,LAB_5)_precedence` $\}$ |
| Relax 1 | $\{$`(LAB_3, LAB_4)_precedence`$\}$ |
| Confl 2 | $\{$`(LAB_1,LAB_2)_precedence,` `(LAB_2,LAB_3)_precedence,` `(LAB_3,LAB_4)_precedence,` `(LAB_4,LAB_5)_precedence,` `MEAL_0_compatible,` `MEAL_1_compatible,` `MEAL_2_compatible,` `MEAL_3_compatible,` `MEAL_4_compatible,` `MEAL_0_timeframe,` `MEAL_1_timeframe,` `MEAL_2_timeframe,` `MEAL_3_timeframe,` `MEAL_4_timeframe`$\}$ |
| Relax 2 | $\{$`(LAB_4, LAB_5)_precedence,` `MEAL_4_timeframe`$\}$ |

Figure 3: Constraint explanations for Example 2.

In this example, relaxations provide the user with minimal ways in which constraints could be changed to fix the schedule. Meanwhile, conflicts give insight into why infeasibility is occurring. For example, `Relex 1` indicates that removing the precedence between laboratory task 3 and 4 would make the schedule feasible. However, `Confl 1` indicates that the precedence between all of the laboratory tasks does not fit in the schedule. A user could use this latter information to alter the original parameters of the schedule rather than having to void an entire task or constraint. One possible solution could be extending the length of the schedule or shortening the length of some of the laboratory tasks, an option which is not revealed by relaxations alone.

This formulation of explainability in terms of conflicts and relaxations also allows a user to ask pointed questions concerning the feasibility of an instance of an ARCPSP problem. Given a feasible instance of a problem, such as Example 1, specific questions may be asked about infeasible modifications of the problem. The modification in Example 2 is gotten by extending the lengths of the laboratory tasks. Hence, the explanation in Figure 3 may be interpreted as an answer to the question: "why can the laboratory tasks not have a duration longer than 120 minutes?"

## 5.3 Implication-Based Enumeration

In the following sections we explore two variations of the subset enumeration algorithm to generate higher-level descriptions of infeasibility. This is accomplished by pushing every constraint to the background and populating the foreground with a fresh set of Boolean variables. Then, a set $L$ of constraints can be constructed that encodes a logical

relationship between the new symbolic variables in the foreground and the actual constraints in the background.

The set $L$ of logical relations linking symbolic variables to the constraints also becomes part of the background. Then, only the set of Boolean variables remains to be enumerated over in the foreground. In practice, this can be accomplished by replacing the Boolean lattice outlined in Section 5 by the symbolic lattice composed of the new variables. This enables the generation of explanations concerning these symbolic variables, which is dependent on the relationship $L$.

Depending on what kinds of constraints populate the foreground, the size of the Boolean lattice which needs to be enumerated over can be greatly reduced. This reduces the number of calls that need to be made to the SMT solver before arriving upon conflicts and relaxation. Additionally, these type of explanations can reduce redundancies and produce more compact descriptions of infeasibility. The following section outlines how this concept can be applied to generate minimal conflicts and relaxations of sets of tasks.

### 5.4 Task Explanations

*Task explanations* can be generated by replacing the foreground (and hence, the power set lattice) with a set of variables representing individual tasks. We introduce a Boolean variable for each task and separate the constraints of the ARCPSP into two classifications: *individual* and *relational*. Constraints (4) and (5) as well as constraints (15) and (16) are individual constraints, involving only a single task. Constraints (6), (7), and (3) are relational constraints, involving multiple tasks. We then encode the constraint that the truth of each task's Boolean variable $J_j$ implies the truth of every one of its individual constraints

$$J_j \implies \mathfrak{I}_j \tag{17}$$

where $\mathfrak{I}_j$ represents a conjunction of the task's timeframe (4), compatibility (5), and feasibility (15, 16) constraints. Visually we can represent the implication as follows, where `LAB_1` represents a Boolean variable and arrows represent logical implications:

```
                    ┌──────────────────────┐
                 ┌─→│   LAB_1_feasibility   │
                 │  └──────────────────────┘
  ┌────────┐     │  ┌──────────────────────┐
  │ LAB_1  │─────┼─→│   LAB_1_timeframe     │
  └────────┘     │  └──────────────────────┘
                 │  ┌──────────────────────┐
                 └─→│   LAB_1_compatible    │
                    └──────────────────────┘
```

For the relational constraints, we add the condition that the truth of all of the dependent tasks' Boolean variables implies its truth. So if relational constraint $C_1$ is between task $J_1$ and $J_2$, then we impose the constraint $J_1 \wedge J_2 \implies C_1$ in the same manner as outlined above:

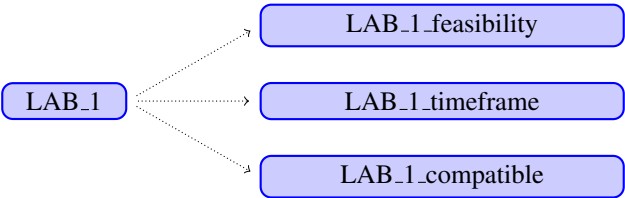

| | Tasks |
|---|---|
| Confl 1 | {LAB_0, LAB_1, LAB_2, LAB_3, LAB_4, LAB_5} |
| Relax 1 | {LAB_1} |
| Confl 2 | {LAB_1, LAB_2, LAB_3, LAB_4, LAB_5, MEAL_0, MEAL_1, MEAL_2, MEAL_3, MEAL_4} |
| Relax 2 | {LAB_5, MEAL_0} |

Figure 4: Task explanations for Example 2.

Hence, individual constraints need only be satisfied if their associated task's Boolean variable is true and relational constraints need only be satisfied if *all* of their associated tasks' Boolean variables are true. Through this logical relationship, we can now enumerate over these Boolean variables, each of which conceptually represents a task. As the Boolean variables are toggled on and off, the associated constraint lattice becomes constrained as if the schedule had been constructed with only that subset of tasks. Executing the enumeration algorithm over this modified foreground for the same over-constrained problem (Example 2) produces a similar set of conflicts and relaxations, part of which is shown in Figure 4. The full explanation includes 3 minimal relaxations and 17 minimal conflicts in total. It took 4.03 seconds to compute the minimal forbidden sets and 0.55 seconds to compute the full explanation.

Here, the task and constraint explanations are quite similar, but this is not always the case. The task explanations can often be much more compact than the constraint explanations because each variable represents many constraints. For similar reasons, the number of total conflicts and relaxations is often greatly reduced. Because of this, task explanations can give more straight forward explanations for the over-constrained problem, but they lack the granularity of the constraint explanations. For example, with the constraint explanations we were able to diagnose that the precedence between the lab tasks was creating an issue rather than a resource or other constraint. The task explanations leave out this detail, sacrificing expressibility for interpretability.

### 5.5 Specification Explanations

In order to draw explanations back to a high-level interpretation of the problem, the foreground can be replaced by a set of human-written specifications. This further reduces the size of the power set lattice that is constructed out the foreground and reduces redundancy in the generated conflicts and relaxations.

Tasks are often formed in groups which share certain scheduling specifications. For example, the meal tasks in Example 1 all share the same parameters except that they are assigned to unique astronauts. When Example 1 was described, such similar tasks were naturally formulated in different categories (e.g., meal, weights, etc.). Hence, specifications for tasks may be more succinctly expressed by making use of these similarities.

An informal, human written list of constraints specifying

| | Specifications |
|---|---|
| Confl 1 | `{Lab tasks must happen in sequence, the scheduling horizon is 6am to 6pm}` |
| Relax 1 | `{Lab tasks must happen in sequence}` |
| Relax 2 | `{The scheduling horizon is 6am to 6pm}` |

Figure 5: Specification explanations for Example 2.

the parameters of Example 1 could be as follows:

– The scheduling horizon is 6am to 6pm.

– Meal tasks must be scheduled between 1pm and 2pm.

– Lab tasks must happen in sequence.

– Each meal/treadmill/weights task must be assigned to a different astronaut.

– Weights/treadmill tasks cannot happen 60 minutes before pre-sleep.

– No more than 1000 units of power may be drawn at once.

– The treadmill tasks require 200 units of power.

– Tasks EVA_SUIT_TEST, HW_GATHER, C-SAT, and SAMP require 400, 400, 500, and 400 units of power.

– There is only one set of weights and treadmill equipment.

Then a relevant logical relationship may be drawn back to the actual set of constraints for each such specification. For example, the precedence relations between the lab tasks could be related through:

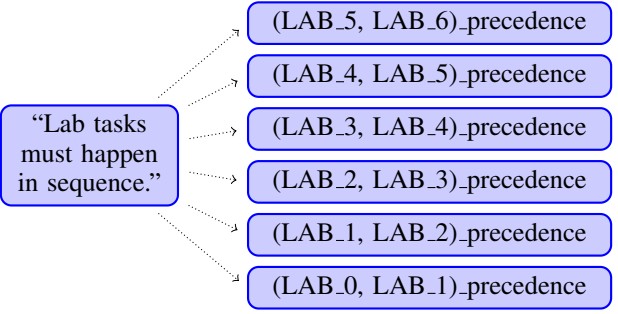

Similar constraints may be encoded for the rest of the specifications, which compose the set $L$ linking the human written specifications to the actual constraints of the problem. This construction allows the generation of *specification explanations*. The specification explanation for Example 1 is displayed in Figure 5. Notice the greatly reduced size of the specification explanation. Unlike the constraint and task explanation, the specification explanation does not suffer from producing many redundant conflicts and relaxations.

A fundamental trade-off exists between the expressibility and interpretability of different kinds of explanations. Low-level explanations involving constraints provide detailed reasons for infeasibility but may be difficult for a human user to parse or understand. In contrast, because the high-level specification explanations correlate directly with the types of constraints which a human planner may think in, they potentially provide more direct and concise information to the user. However, they lack the fine tuned granularity of information that constraint and task explanations provide. For example, if only a single precedence constraint between the laboratory tasks was causing an issue, the specification explanation would obscure which of the constraints is responsible.

## 6   Conclusion

We introduced the agent resource-constrained project scheduling problem (ARCPSP) along with an associated difference logic encoding. We proposed a general framework for generating minimal conflicts and minimal relaxations based on the MARCO algorithm and demonstrated how it could be used to generate varying types of descriptions for why infeasibility is occurring in instances of the ARCPSP. The framework outlined in this paper is general enough to be applied to constraint satisfaction formulations for various other scheduling and planning problems. These ideas may potentially be further extended to different kinds of formal languages, such as linear temporal logic, that are used to describe planning problems.

### 6.1   Future Work

In an interactive system, such as a scheduling software, when a user attempts to make an infeasible modification, it may be useful to generate a reason for the infeasibility in real time. Similarly, a user could query whether a modification to a feasible schedule would preserve feasibility and, if not, why not? Explanations similar to the ones constructed throughout this paper may likely be used to such an effect. Investigating methods for synthesizing natural language sentences out of the explanations is also subject to future research.

Following the goal of QuickXplain (Junker 2004), given a partial ordering of constraint or task importance, preferred conflicts and relaxations may be explored earlier and full explanations may list conflicts and relaxations in preferential order. Such functionality would be especially useful in cases for which generating the full explanation is intractable. A preferential ordering of explanations may be achieved by adding and removing constraints during the **Grow** and **Shrink** steps based on the constraint preference ordering. Similarly, methods for enumerating disjoint (or otherwise distinct) conflicts may also be useful for producing a representative set of conflicts as concisely as possible.

Currently, the most limiting bottleneck for scaling to larger problem instances comes from the number of minimal forbidden sets which can grow exponentially with the number of tasks. Certain lazy clause generation algorithms (Laborie 2003) may be used to represent resource constraints in a more efficient manner. Such representations may also be adapted to implement *consumable* resources in an explainability setting.

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
