# OpenReview forum: "Human-Understandable Explanations of Infeasibility for Resource-Constrained Scheduling Problems"
_icaps-conference.org/ICAPS/2019/Workshop/XAIP — XAIP 2019_

### Official Review · AnonReviewer2 · 2019-05-09
**Strong paper on explaining plan/schedule infeasiblity**

**Rating:** 4
**Confidence:** 2

**Review:**

This is a well-written, clear, and relevant paper to the conference.  The paper introduces a framework for explaining scheduling and planning failures by generating minimal conflicts and relaxations. The work is well motivated by a description of planners at NASA and is used as a running example throughout the paper. The problem of explaining why a plan fails is a core focus of this workshop thus this paper will be highly relevant to the audience.

Small comments / grammar:
	* 6.1: "natural language sentences out the explanations" - "out of"?

---

### Official Review · AnonReviewer1 · 2019-05-13
**Towards an approach for Explainable Scheduling**

**Rating:** 4
**Confidence:** 3

**Review:**

This paper considers the problem of explaining while a schedule is not feasible.
The work focuses on resource-constrained project scheduling problem, and the approach is based on using SMT for identifying minimal conflicts and minimal relaxations that will form the explanations of why a schedule is not feasible.

The work is clearly relevant to the workshop.

I have a few concerns, though.
First, it is not clear what the form of the explanations will be. While the authors do show an example, it is not clear how this can generalise.

Second, the effectiveness of the proposed explanations still needs to be assessed and properly evaluated.

Third, even from a computational point of view, it is not clear (nor clearly evaluated) how the approach can scale to larger/different problems.

Finally, the authors should comment on the similarities/differences with
"Generating Explanations for Mathematical Optimisation: Solution Framework and Case Study"
Christina Burt, Katerina Klimova, and Bernhard Primas. Proceedings of XAIP 2018.

---

### Decision · Program_Chairs · 2019-05-15

**Decision:**

Accept

**Comment:**

The reviewers agree to accept. Please address all review criticism as best possible for the final paper version and its presentation at the workshop. Looking forward to discuss your work at the workshop!